# Recovering a lost seismic disaster. The destruction of El Castillejo and the discovery of the earliest historic earthquake affecting the Granada region (Spain)

Paolo Forlin[1,2]*, Klaus Reicherter[3], Christopher M. Gerrard[2], Ian Bailiff[2], Alberto García Porras[4]*

1 Alma Mater Studiorum, University of Bologna, Bologna, Italy, 2 Durham University, Durham, United Kingdom, 3 RWTH Aachen University, Aachen, Germany, 4 University of Granada, Granada, Spain

* paolo.forlin@unibo.it (PF); agporras@ugr.es (AGP)

**Data Availability Statement:** All relevant data are within the manuscript and its Supporting information files.

## Abstract

This paper discusses recent archaeological fieldwork conducted at El Castillejo, a medieval Islamic settlement in Los Guájares, Granada, southern Spain. Results from combined archaeological excavation and archaeoseismological assessment of standing structures suggest that the site was affected by a destructive earthquake during its occupation. Radio-carbon samples and OSL analysis point to a seismic event in the period CE 1224–1266. The earthquake occurred within an area marked by a 'seismological gap' in terms of historic seismicity and the causative fault has been tentatively identified in the Nigüelas-Padul Fault System which lies north of the settlement. This event is not recorded by national or European seismic catalogues and represents the oldest historic earthquake in the Granada area. Our work stresses the significant impact that targeted archaeological investigations can generate in our understanding of the local historic seismicity, thus providing clear implications for seismic disaster prevention and reduction.

## 1. Introduction

Catalogues of earthquakes rely heavily on archival sources, so historic seismic events in regions of the world with little surviving written documentation can be severely under-represented [1–4]. Archaeology offers an alternative avenue of investigation but tends to play an ancillary role at best in historic seismology. Here we argue that recent developments in 'archaeoseismology' or 'earthquake archaeology' have something new to offer through more rapid reconnaissance techniques [5–8], 3D recording and modelling [9, 10] coupled with the use of geoarchaeology [11] to evaluate both above- and below-ground deposits [12, 13]. Nevertheless, architectural analysis is rarely paired with the investigation of buried stratigraphy and recent work at El Castillejo (Los Guájares), an Islamic hilltop settlement near Granada (Spain), highlights the potential of this approach. In particular, by combining (i) the assessment of seismic damage on ruined but standing buildings, with (ii) targeted excavation, and (iii) absolute dating methods,

**Funding:** PF was awarded a Marie Skłodowska Curie fellowship called ArMedEa ('The archaeology of Earthquakes in medieval Europe', grant n.626659) and a British Academy small grant (SRG/171316). The funders had no role in study design, data collection and analysis, decision to publish, or preparation of the manuscript.

**Competing interests:** The authors have declared that no competing interests exist.

we show how archaeology is well positioned to make a fuller contribution to our understanding of past seismicity and contemporary seismic hazard.

## 1.1. El Castillejo in its geographical and tectonic setting

The site of El Castillejo in the Las Guájaras sierra overlooks the Toba valley in the southernmost sector of the central Betic Cordilleras in southern Spain [14, 15] (Fig 1A). This part of Andalusia is among the most seismically-active areas in Europe and characterised by high energy destructive earthquakes [16–18]. A diffuse plate boundary at the collision zone of the Eurasian and Nubian plates induces mainly shallow earthquakes with a maximum horizontal stress ($S_{Hmax}$) oriented NW-SE. This stress regime developed after the collision and formation of the Betic Cordilleras in the late Miocene, together with an extension running NE-SW and Neogene basin formation [19–21]. These intra-montane basins are fault-bound and seismically evidenced by frequent micro-earthquake activity [18, 22]. The last major seismic event in the Granada area was the Arenas del Rey earthquake in 1884, which affected 14,000 km$^2$ and destroyed a number of settlements, claiming nearly 1,000 lives [23–25]. However, the seismic historic record is rather limited for this region and no earthquakes are recorded before the 15th century [26, 27].

In the southern part of this region lies El Castillejo, a fortified hilltop village enclosed by an elliptical circuit of walls marked by rectangular towers and a 24m-long gatehouse [15, 28] (Fig 1B; for plan see Fig 3). With an internal area of approximately 3,500 m$^2$, the village was once densely occupied with houses spread across both the northern and southern slopes of the hill.

The Islamic houses at the site are laid out around a central patio on at least two floors [29], common layouts for housing in medieval al-Andalus [30]. While the larger houses (e.g. buildings 8 and 10) cover c.75 m$^2$ and consist of three rooms in a U-shaped plan around a patio, others (e.g. buildings 2 and 3) occupy more modest plots of c.45 m$^2$ and are L-shaped in layout with a smaller patio. Granaries, stables and other communal facilities can also be identified, including a cistern (building 9) and the remains of a hydraulic infrastructure c.100 m outside the western entrance [31]. All of these structures were constructed using rammed earth (locally known as *tapia* or *tapial*), a construction technique with a significant legacy in Spain. Rammed earth is made compacting soil dug from the ground within removable formwork [32]. Rectangular blocks or 'lifts' of rammed earth rest one upon another without the use of mortar, often with the putlog holes from the formwork being left visible.

## 2. Methods

In order to assess potential seismic destruction of the settlement and the chronology of the seismic event, we combined standing wall analysis, targeted stratigraphic excavation, and absolute dating methods (OSL and radiocarbon dating). This archaeological study of the site was paired with a palaeoseismological analysis of the area where El Castillejo is situated.

## 2.1. Structural damage assessment and stratigraphical excavation

We systematically recorded structural damage on standing walls and other structures in the field. Damage was identified, mapped, photographed and catalogued. In order to map the damage, a new plan of the site was created using a photogrammetric drone survey. Structure-from-motion documentation was also employed to characterise, measure, and quantify seismic damage on a 3D model of Building 4 [10]. Archaeological excavations then documented all the buried deposits stratigraphically and we used structure-from-motion documentation of the trenches to produce archaeological plans and sections, whose location was topographically recorded on site with a total station. Archaeological finds were systematically analysed,

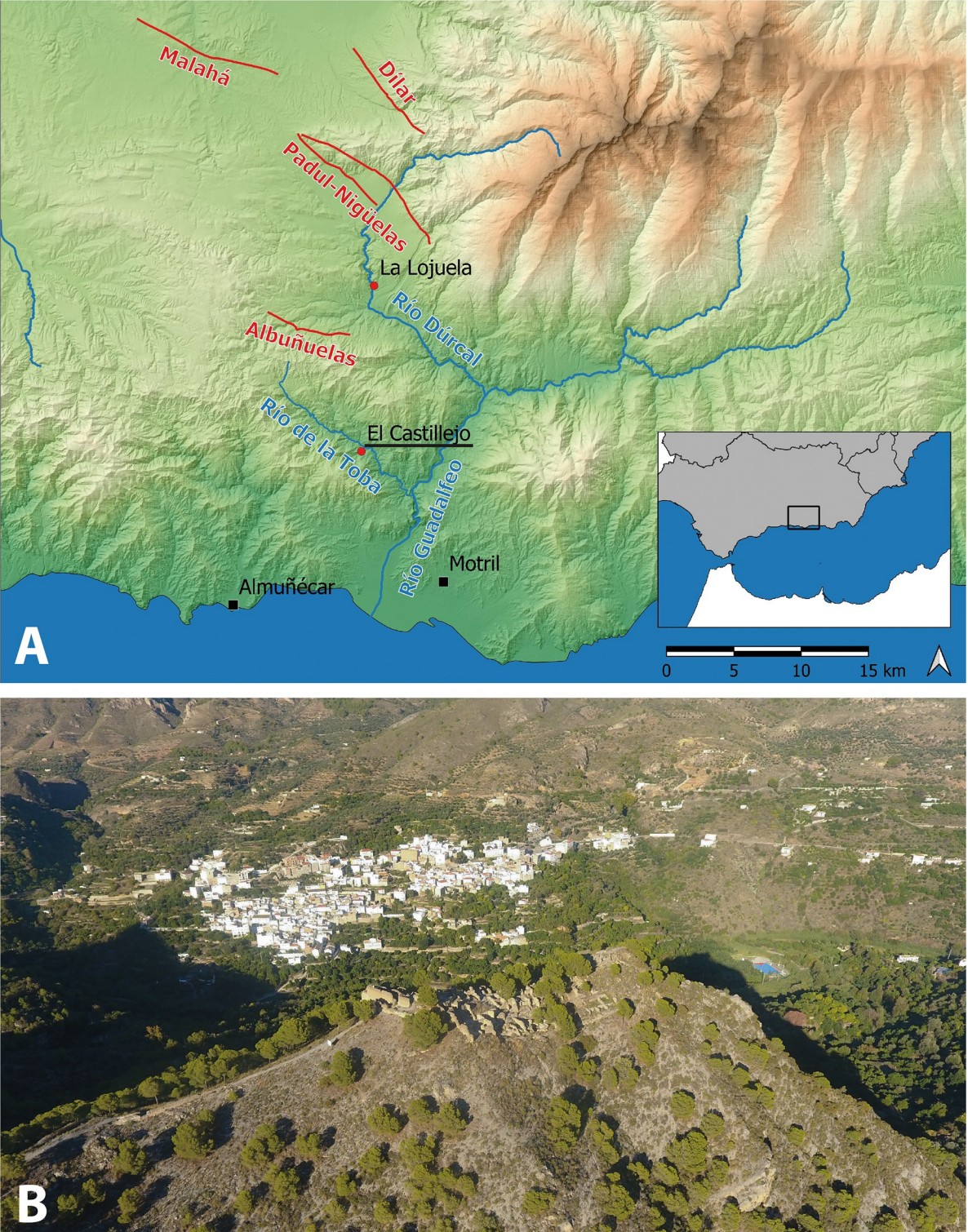

**Fig 1.** A: Geographical location of El Castillejo (map generated by the authors using QGis 3.26; Basemap: MDS05 2020 CC-BY 4.0 ign.es; seismic faults from the QAFI database [17]); B: Drone view of the archaeological site from SW. The village of Guájar Faragüit lies in the background. Photograph by P. Forlin.

photographed, catalogued and stored in the Department of Medieval History and Historio-graphical Sciences and Techniques of Granada University.

## 2.2. OSL

Optically stimulated luminescence (OSL) techniques were applied to samples of sediment and fired ceramic brick from five locations with the objective of dating the deposition and burial process in the case of sediment and manufacture in the case of the brick [33]. For both sediment and brick samples coarse grain quartz samples were extracted using conventional preparation procedures for luminescence measurements [33]. As initial testing revealed that none of the sediment samples contained quartz producing measurable OSL signals, further measurements with these samples were suspended. However, the ceramic materials yielded quartz coarse grains with characteristics suitable for dating measurements and a summary of the results obtained for the two bricks sampled is reported here. Further information about the OSL dating is provided in the Supplementary material.

## 2.3. Radiocarbon dating

The samples were radiocarbon dated by the SUERC Radiocarbon Laboratory, University of Glasgow. Results were later processed with Bayesian Statistics. All calculations were performed in OxCal 4.4.4 and using IntCal20 [34].

## 2.4. PalaeoShake maps

The archaeological evidence stimulated a multidisciplinary study involving palaeoseismological analysis. In addition to field observations, we calculated scenario-based ground-shaking distribution and intensity, i.e. a so-called scenario "Palaeo"Shakemap [35]. Different scenarios with varying values of earthquake magnitude and depth were determined. Input parameters included nearby seismogenic faults like the Ventas de Zafarraya Fault, the Nigüelas and Padul Faults that could have affected the El Castillejo site. Based on the USGS ShakeMap manual [35], a Phyton script was used for the calculations. With input parameters of the magnitude of the earthquake, hypocentral depth, fault strike and coordinates, earthquake-generated shake distribution and EMS intensity [36] were estimated, originating from a 2D-source (scenario hypocentre). A high-resolution Digital Elevation Model (DEM) was first needed to extract information on slope and elevation. Second, shake radius and intensity were assessed based on the DEM and input data about magnitude, fault strike and depth. Side corrections, e.g., distance correction, elevation differences and absorption coefficient, were also carried out.

## 3. Results

### 3.1. Previous archaeological investigations (1985–1989; 2015)

El Castillejo was extensively excavated in the period 1985–1989 and then in 2015 for a total of five archaeological campaigns. The overall area investigated is 1,800 square metres, corresponding to around 40% of the overall surface of the site [37–39]. The pottery, which included more than 400 reconstructable vessels, suggests that settlement began in the 11th-12th century CE. Two occupation phases were identified. The end of the first phase was stratigraphically marked by substantial destruction layers including fallen blocks of rammed earth walls, large quantities of plaster fragments and roof tile which had sealed both smashed and intact pottery containers which had been left *in situ*. In their original interpretation, the excavators understood these features to be the result of prolonged degradation and erosion caused by a gradual process of collapse of the buildings [37]. In the absence of absolute dates, the chronology for

the end of the first phase was not assessed. In contrast to the eastern part of the site which was entirely abandoned, in the western part of El Castillejo where the buildings are today better preserved and the walls still stand to a height of c.2m, the archaeological investigations identified a later phase of reoccupation which can be seen in the standing architectural remains. The fortified gatehouse here, together with a few residential buildings nearby, appears to have undergone a process of controlled demolition and later rebuild using a newly prepared but lower-quality rammed earth together with large chunks of rammed earth rubble which had been salvaged from the collapsed buildings (see Fig 2F and 2G). This phase of repair must have taken place within a 'semi-abandoned context' [38] of the site as a whole. The date for this reoccupation is unclear (see in particular [39]), though the pottery evidence suggests the final abandonment of the site before or around the beginning of the 14th century [15]. Nevertheless, sporadic occupation is also attested in the 16th century, when resistant Moriscos groups possibly used the ruined site as a refuge [40].

In summary, the settlement history of El Castillejo was understood to be marked by a dramatic episode of change between its first and second phases. However, the potentially sudden and devastating destruction of the site at the end of Phase 1 was never fully investigated (see in this regard Malpica et al. [38]) and, although the excavation unearthed some substantial evidence for possible seismic damage, destruction by earthquake was not openly considered.

## 3.2. Assessment of structural damage and deformation on standing walls

In our reconsideration of the evidence, structural damage and deformation on standing walls unearthed by previous excavations were first catalogued and mapped across the site. Among the features identified were:

- **Penetrative fractures**. Horizontal, vertical and shear cracks had passed right through the full width of the rammed earth walls (usually around 45 cm) producing symmetrical fractures on either side. Fractures like these were clustered in the western half of the village where the majority of the standing structures are still preserved today (buildings 0, 2, 3, 4, 5, 6, 7, 8, 10, and 40; see examples in Fig 2A and 2B).

- **Displaced or shifted blocks of walling**. Displacements in which blocks of rammed earth become vertically misaligned were identified in buildings 2, 3 and 4 in the north-east part of the site and measure between 5 and 10 cm (Fig 2C).

- **Tilted walls**. Tilting was observed at the bases of four (partially collapsed) walls in buildings 4, 9 and 10 as well as at the base of the northern outer wall of the settlement (Fig 2D).

- **Fractured and misaligned walls**. Lateral seismic impacts can fracture and then misalign a structure horizontally. This effect was identified in building 30 and possibly along the aqueduct outside the village (examples in Fig 2E).

- **Collapsed walls**. The collapse of buildings is particularly clear in the eastern part of the site, where standing walls are essentially absent today. In this area, for example Building 30 was completely destroyed down to its foundations and a spread of destruction rubble is still visible across nearby buildings. As mentioned above, blocks of fallen walls were a distinctive feature in the archaeological record unearthed by previous excavations.

Fig 3 is a map of all the structural damage detected at El Castillejo. Notably, structural damage is exclusively observed on first phase structures and absent on second phase structures which, as we have already noted, also reused construction materials from older damaged buildings (see for instance Fig 2F and 2G). In itself, this strongly suggests that a destructive event

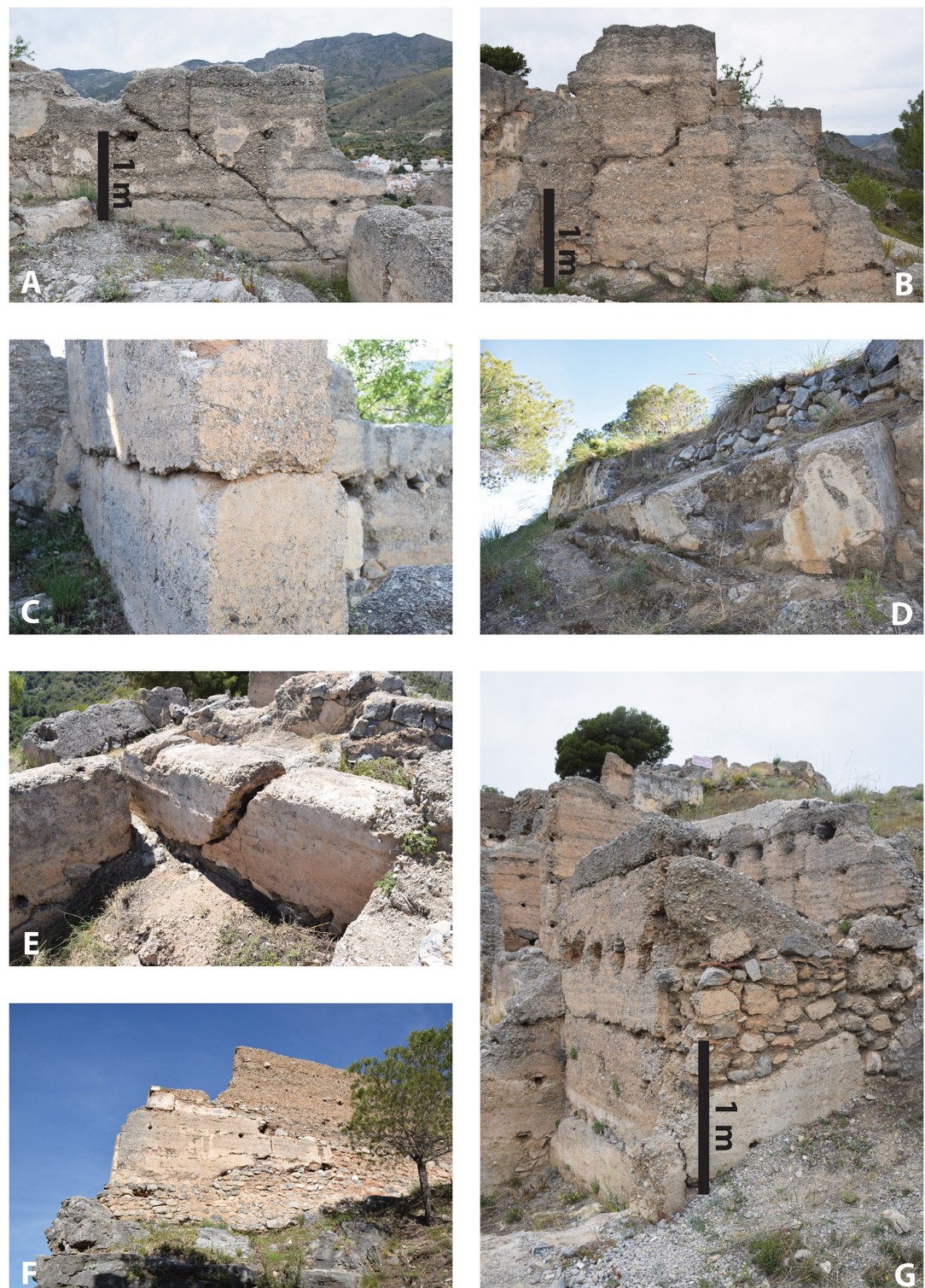

**Fig 2.** Examples of seismic damage and restoration observed at El Castillejo: A-B: shear cracks in Building 4; C: horizontally-shifted block of rammed earth in Building 3; D: tilted block of the outer wall; E: broken and displaced wall in Building 30; F: post-earthquake restoration of the western gatehouse and G: Building 10. Photographs by P. Forlin.

terminated the first phase of occupation, although the precise chronology of this episode could not be resolved without recourse to further archaeological investigation of the buried stratigraphy.

### 3.3. Targeted excavation (2018)

Four trenches were excavated, of which three are discussed below (Fig 3 for location).

**Trench 1** (2x3m) was located at the centre of the site where a tilted wall and a number of rammed earth blocks were visible at the surface (Fig 4). The excavation identified the eastern limit of a building (tilted wall 115, aligned N-S) with an internal room (probably a patio) delimited by two perpendicular walls, 102 (E-W) and 107 (N-S). Parallel N-S walls 107 and 115 probably define a narrow corridor, wall 107 having chevrons incised on its eastern plaster face. Wall 102 served as a retaining wall for a terraced area which had been levelled with gravel and schist clasts (117). Taken together, these features correspond to the construction and occupation of a building which had been terraced into the hillside by modifying the original profile of the bedrock (118). The occupation surfaces of this (probably domestic) building consisted of a coarse gravel layer 116 (sitting on 117 north of wall 102) and two plaster floors. These sat directly on the bedrock in the southwestern (109) and southeastern quadrants (122) of the trench.

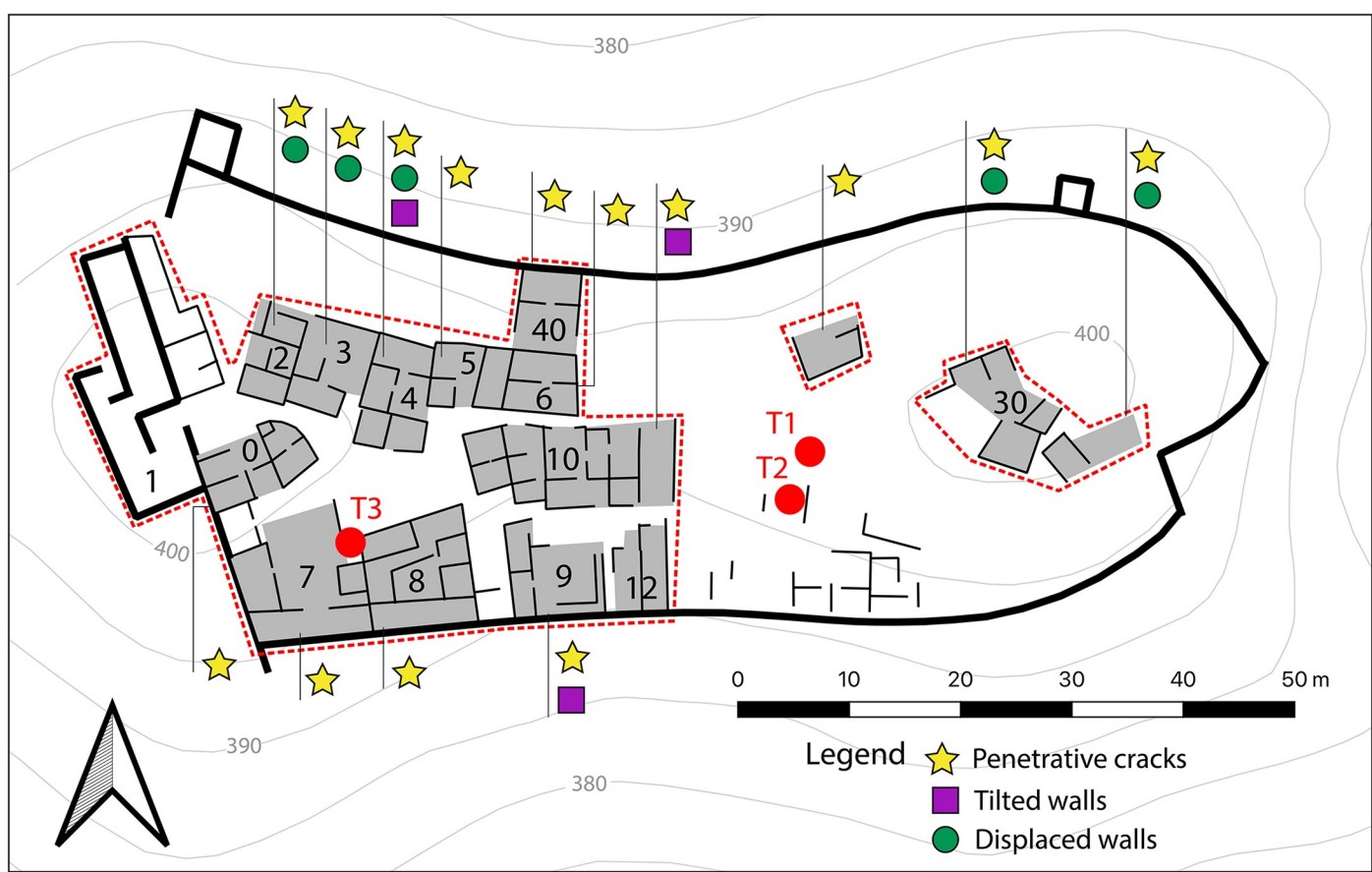

**Fig 3. The distribution of structural deformation and damage grouped by buildings (see key for the types of damage).** The dot red lines indicate previously excavated areas. Trenches discussed here are shown as red circles (map generated by P. Forlin using QGis 3.26).

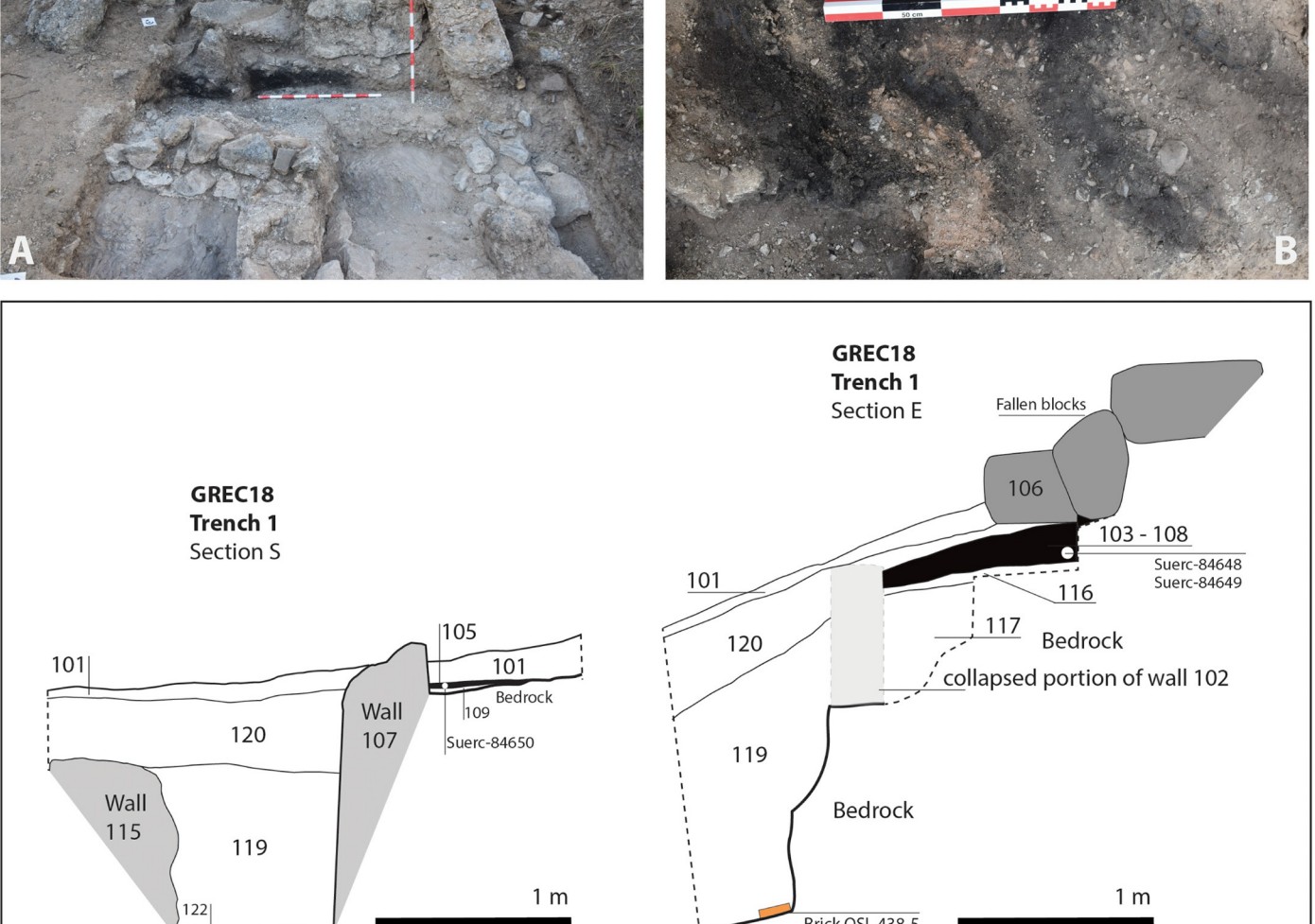

**Fig 4.** A: Trench 1 under excavation. Note the burning layers 103–108 sealed by debris and rubble 106 and 120; B: Close-up of the in situ carbonised beams; C: Stratigraphic section S; D: Stratigraphic section E.

*In-situ* layers of burning (103 and 108) were discovered both north and south of the wall delimiting the possible internal patio. Four parallel beams had burnt *in situ* and, within the internal room, the burning layer (105) was intermixed with roof tiles, quantities of nails, worked iron and copper artefacts possibly derived from a wooden box or chest. These areas of *in situ* burning, along with the internal walls 102 and 107, were all found to be sealed by substantial blocks of fallen wall (106) and smaller fragments of loose, decayed rammed earth (101, 110, 119, 120). Given that this clay lump debris was reddened through contact with the burnt timbers, it can be inferred that the fire and the collapse of the building were a single contemporary event. The soil horizon above (101) indicates that this part of the site was subsequently abandoned.

**Trench 2** (2x2m) flanked an almost completely buried standing wall about 8 m downhill to the south (Fig 5). The earliest occupation here also corresponded with the construction and occupation of a rammed earth building. The building's external N-S wall (206) was the eastern limit of the trench, but an internal N-S wall (205) was also identified, together with an internal mortar floor (211) which had been laid onto a stepped surface running uphill towards Trench 1. The internal wall (205) divided two compacted occupation surfaces (208 to the east; 210 to the west) with pottery sherds lying directly on the bedrock (209).

A rapid onset event had also led to the collapse of this building. Occupation surfaces 208 and 210 were sealed by broken slabs of rammed earth (204/207), interpreted as the remains of smashed walling. Both 204 and 207 contained *in-situ* broken pottery and broken roof tile. The interior of the building was completely infilled by large blocks of rammed earth mixed at higher levels with a thick layer of decayed and loose rammed earth (202). Voids between the fallen blocks of walling had allowed finer crumbs of sediment to percolate downwards, probably derived from the decay of the rammed earth at higher levels which had filtered down through empty spaces over time. Finally, a colluvium of loose and degraded rammed earth (201) and topsoil (200) covered this part of the site and represents the subsequent abandonment of the area.

**Trench 3** corresponded to an exposed section in the external space flanked by building 7 to the west and building 8 to the east (Fig 6; see Fig 3 for location). The section shows an original plaster floor at the base which was later re-surfaced with schist gravel (318). In this case, a thick deposition of laminated sediments lay above the gravel surface (316, 313). These were interpreted as gravitational run-off deposits in an area which had not been maintained. A small triangular wedge of degraded *tapia* (314), possibly due to the erosion (weathering) of wall 321, was recorded in 313. At the top of this sequence, a possible stasis in the formation process of the laminated sequence is represented by 312 and 311, two dark organic layers which might be occupation surfaces. In a further phase, the eastern wall of building 7 (320) was built. The upper part of the sequence described above is cut by its foundation trench 308 and filled by 307. A possible occupation surface (305 = 306) developed in association with this wall.

Surface 305–306 was completely covered by a thick layer of debris including substantial rammed earth blocks and loose, decayed rammed earth (304). Layer 303 above contained a cluster of large fragments of roof tile and some rammed earth rubble with a developing soil horizon in the upper part of the section. The abandonment of this site's sector is indicated by a thick layer of colluvium (302, lower part; 301, upper part) and topsoil (300).

Overall, no evidence for reoccupation was found in our trenches, the debris layer sealed by colluvial deposits confirms the abandonment of the excavated areas. This is consistent with what was observed by previous investigations which, as highlighted above, stressed that restoration and repair were limited to the outer wall and a few first-phase buildings located exclusively in the western part of the village.

## 3.4. Dating the destructive event

The dating of the destructive event was acquired by considering pottery typologies, radiocarbon dates from remains of short-lived plants, and OSL dates from manufactured bricks.

**3.4.1. Pottery.** The pottery assemblage recovered in 2018 was essentially domestic and functional in character. Like the ceramics recovered from previous excavations [15], most vessels were related either to the storage of foodstuffs and water (*jarras*), collective consumption of food (*ataifores, jarritas*) or the preparation and cooking of food (*cazuelas, marmitas*). One almost complete basin (*lebrillo*) from Trench 2 had been exposed to heat and could have been

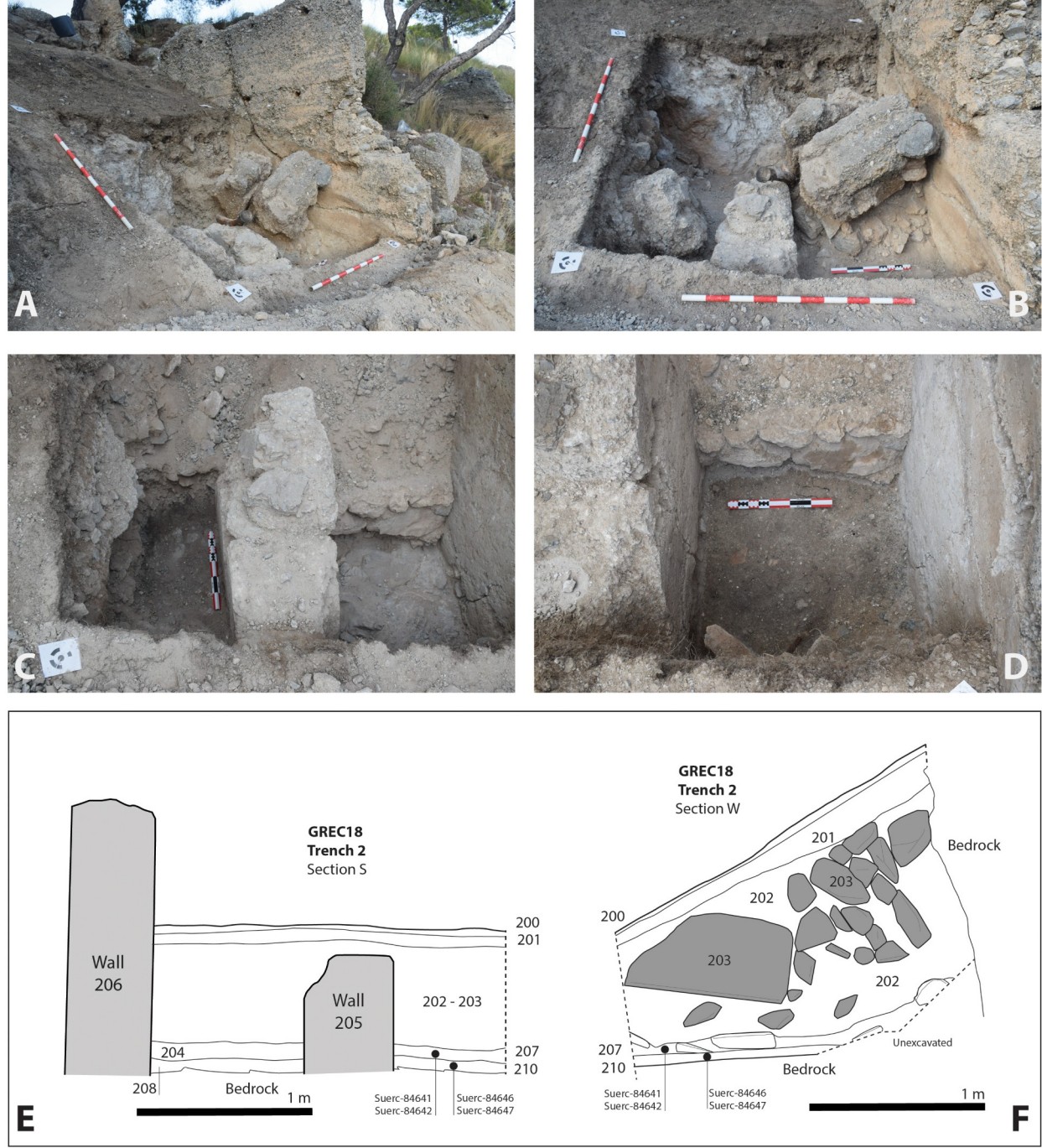

**Fig 5.** A: Trench 2 under excavation looking NE. Note wall 206 showing penetrative fractures associated with fallen blocks of rammed earth; B: Close up of debris and fallen blocks 202 and 203; C: the occupation surface 208 looking north; D: the occupation surface 210 looking north; E: Stratigraphic section S; F: Stratigraphic section W.

used as a brazier or a small 'portable' hearth. All these finds are consistent with a 13th-14th century chronology.

**3.4.2. Radiocarbon dating.** Eight samples of the remains of short-lived plants recovered from contexts directly sealed by the destruction layers were submitted for radiocarbon dating

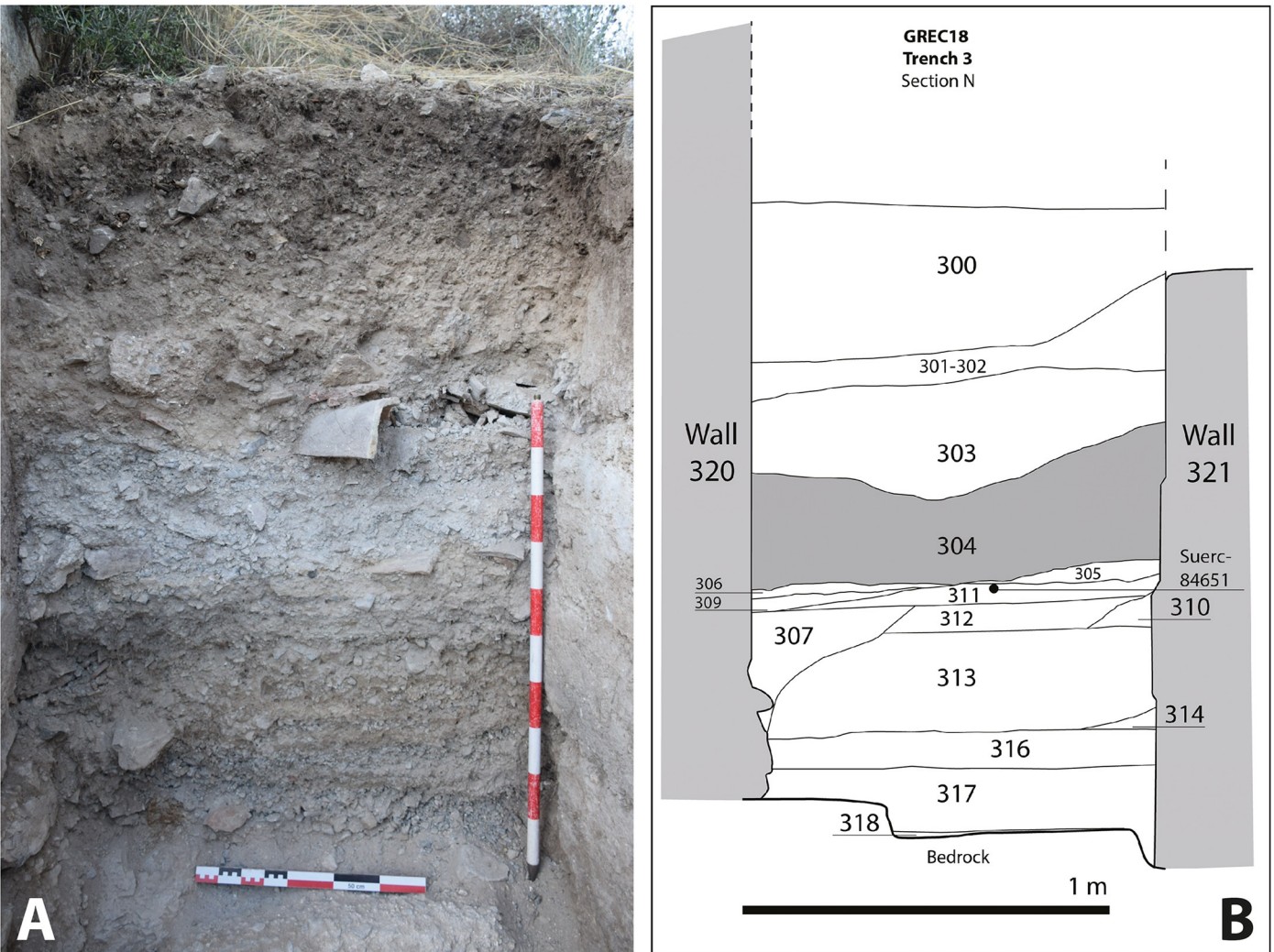

**Fig 6.** A: Trench 3 looking N; B: Stratigraphic section. 304 corresponds with the rammed earth block and loose horizon.

(see sections in Figs 4–6 for the stratigraphic position of the selected samples). This organic material, which was found on the occupation surfaces, had either been charred in the fire which followed the destructive event (in Trench 1: contexts 105 and 108) or had been deposited immediately beforehand (Trench 2: contexts 207 and 210; Trench 3: context 3011). The individual dates obtained consistently point to an event in the 13th century CE. As they relate to the same stratigraphic event, they can be combined to produce a more precise calendar age estimate. All calculations were performed in OxCal 4.4.4 and using IntCal20. Dates were combined separately for each trench and as a single dataset (Table 1; Fig 7). The results show no differences between the trenches and indicate a date for the earthquake within a time interval of ca. 40 years between CE 1224–1266 at 95.4% probability. Radiocarbon dating extracted from the sealed surfaces thereby provides an entirely consistent chronology for the destruction of the site.

**3.4.3. OSL dating.** OSL dating results are reported in Table 2. One brick (sample 438–5) recovered from the surface sealed by layer 119 in Trench 1 (Fig 4) produced an OSL date of CE

**Table 1. Radiocarbon dates and Bayesian statistics.** Data provided by the SUERC radiocarbon laboratory of the University of Glasgow.

| Trench | Context | Sample | Laboratory code | Material | δ¹³C (‰ VPDB) | Conventional radiocarbon age | | Combined and calibrated 68.2% probability range (years AD) | Combined and calibrated 95.4% probability range (years AD) |
|---|---|---|---|---|---|---|---|---|---|
| Trench 1 | 105 | C14-07 | SUERC-84650 (GU50193) | Charred fruit fragment cf. *Vitis vinifera* L. | -23.8 | 784 | ±34 | 1228–1265 | 1222–1270 |
| | 108 | C14-05 | SUERC-84648 (GU50191) | Charred fragment of *Olea europaea* L. | -22.6 | 841 | ±34 | | |
| | 108 | C14-06 | SUERC-84649 (GU50192) | Charred *Triticum* sp. | -23.9 | 761 | ±34 | | |
| Trench 2 | 207 | C14-01 | SUERC-84641 (GU50187) | Charcoal: *Pinus* sp. | -25.3 | 782 | ±34 | 1227–1261 | 1222–1267 |
| | 207 | C14-02 | SUERC-84642 (GU50188) | Charcoal: *Pinus* sp. | -25.0 | 800 | ±34 | | |
| | 210 | C14-03 | SUERC-84646 (GU50189) | Charred *Hordeum* sp. | -23.9 | 816 | ±34 | | |
| | 210 | C14-04 | SUERC-84647 (GU50190) | Charred *Hordeum* sp. | -25.3 | 811 | ±34 | | |
| Trench 3 | 311 | C14-08 | SUERC-84651 (GU50194) | Charcoal: Fabaceae (roundwood) | -24.5 | 789 | ±34 | 1227–1269 | 1180–1281 |
| All | | | | | | | | 1228–1263 | **1224–1266** |

1300±55, indicating manufacture within the chronological range suggested by the radiocarbon dates. However, given the evidence for burning in the overlying deposits, a temperature of 300˚C would be sufficient to 'reset' the luminescence clock and the OSL date calculated consequently corresponding to the last heating event. The overlap of the radiocarbon and luminescence (95% probability range) age ranges (Fig 7) suggests that this is plausible and that the event dated by OSL is related to the conflagration. The second brick tested (sample 438–2; wall 1003) was embedded in a first phase rammed earth wall of building 10. If it is assumed that the brick used in constructing the wall had been recently manufactured (Use History A), the OSL date obtained (CE 945±70) places its manufacture during the c.10th-11th centuries CE, and thus providing an estimate of the construction date of the building. However, the possibility that the brick had been reclaimed from an older brick structure should also be considered (Use History B), and in that case a slightly younger age is obtained (CE 1005±65; see S5 File). Whether the brick was recently manufactured when included in the original build of the wall or reclaimed from an older structure and later inserted here is not known [41].

## 4. Discussion

### 4.1. The earthquake that destroyed El Castillejo

Based on the archaeological evidence presented above, we suggest that El Castillejo provides vivid physical evidence for the impact of a major earthquake. Tilted, displaced and collapsed walls, penetrative fractures in standing structures, collapsed floors and roofs, broken *in situ* domestic objects, evidence of fire, all these are characteristic of severe seismic shaking. **Penetrative fractures** recorded on standing walls unearthed both by previous excavations and found by our fieldwork (such as cracked wall 206 in Trench 2) correspond to in-plane or out-of-plane cracking which is entirely consistent with seismic damage. This kind of damage has previously been observed in both archaeological parallels and contemporary scenarios such as the recent 2023 Oukaïmedene Morocco Earthquake [3, 42–45], but also emerges as a result of the simulated seismic loading of rammed earth structures [46–52]. **Deformation along lift interfaces** as observed in the displaced blocks of walling is a well-known behaviour of rammed earth structures affected by earthquakes [52] and can be especially distinctive in rammed earth buildings because mortar is not used to cement the lifts together. This characteristic facilitates the horizontal sliding of rammed earth blocks. When floor beams and roof timbers are displaced, they can also push and strike at the walls, either causing or exacerbating lateral wall movement. **Tilting, misalignment and the sudden collapse of walls** (as identified in

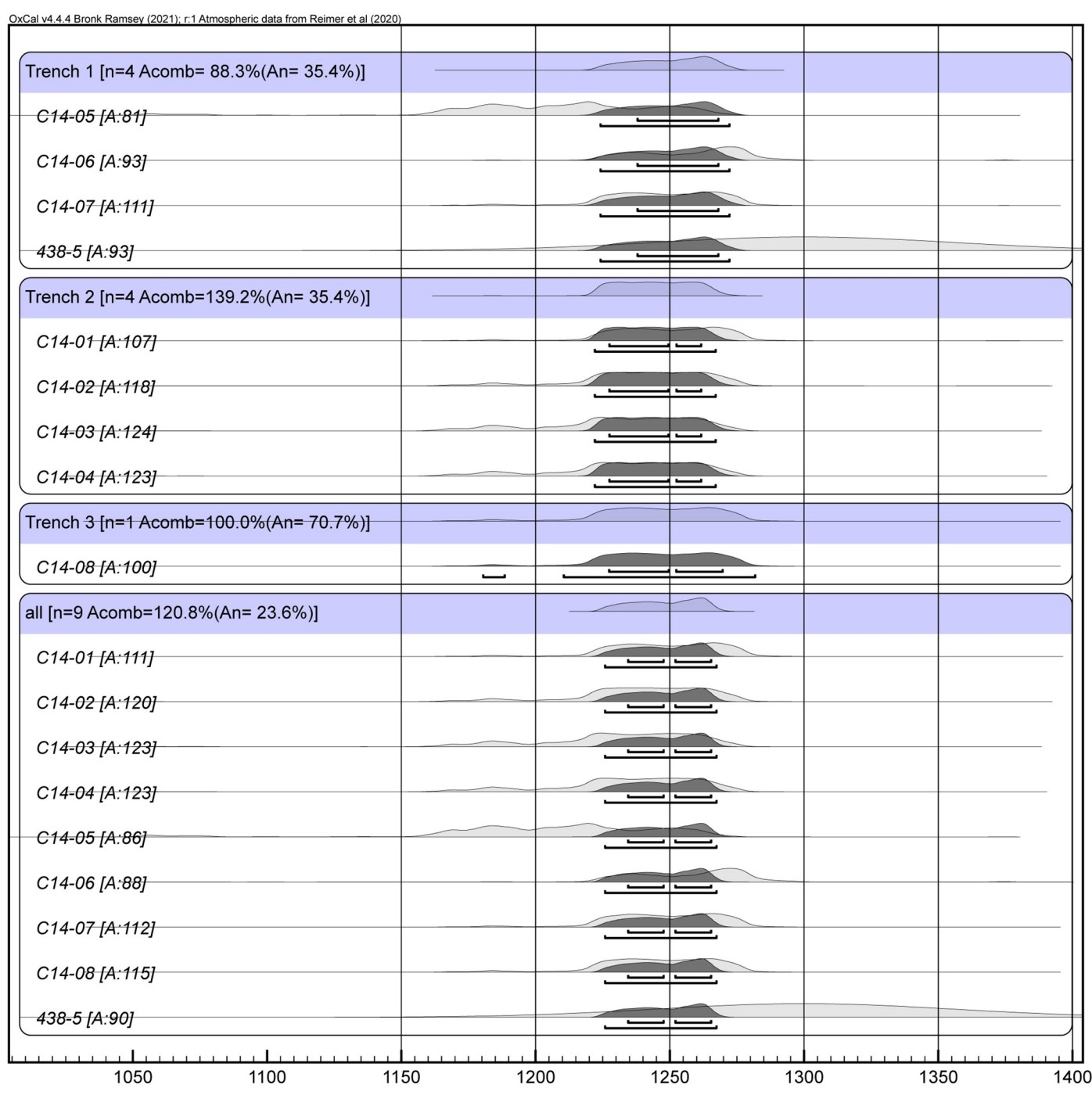

**Fig 7. Plot of the radiocarbon dates obtained from the analysed samples (image by G. Capuzzo).**

**Table 2. OSL dating.** The uncertainty associated with the luminescence age (test year, 2019) is given at the 68% level of confidence, calculated following a procedure of error propagation [41]. The luminescence age corresponds to the quotient of the paleodose, P, and the dose rate, $D_r$; those shown were rounded to the nearest 5 years. The two ages calculated for sample 438–2 take account of different histories of use of the brick following manufacture (A and B), as discussed in the main text. The overdispersion in P, $\sigma_B$, is an indicator of beta dose rate heterogeneity in the case of heated samples, and the number of determinations (aliquots), n, is given in the adjacent column. In calculating the annual dose rate, $D_r$, the beta dose rate $D_\beta$ includes a small contribution (0.035 mGy/a) from radiation emitted by lithogenic radionuclides within the quartz grains. The average moisture content of ceramic and environmental materials was assumed to be 3±1% for sample 348–2 and 10±2% for sample 348–5, by weight.

| Lab code | Context | P | $\sigma_B$ | n | Use History | $D_r$ | $D_\beta$ | $D_{\gamma+cos}$ | Age ±1σ | Date ±1σ |
|---|---|---|---|---|---|---|---|---|---|---|
| | | Gy | | | | mGy/a | Frac | Frac | a | CE |
| 438–2 | 1003 | 4.66±0.10 | 0.09 | 21 | A | 4.30±0.11 | 0.78 | 0.22 | 1075±70 | 945±70 |
| | | | | | B | 4.58±0.14 | 0.73 | 0.27 | 1015±65 | 1005±65 |
| 438–5 | 119 | 2.10±0.08 | 0.12 | 13 | | 3.01±0.08 | 0.74 | 0.26 | 720±55 | 1300±55 |

destruction layers 106, 119, 120, 202–203, 204, 207 and 304) are other well documented clues to seismic destruction and widely observed in earthquake-affected archaeological sites and contemporary episodes [42, 53, 54]. **Domestic fire**, as observed in layers 105 and 108, is another co-seismic secondary effect documented elsewhere, for instance in the churches of Venice in 1117 [1] or the city of Basel in 1356 [53], to name two medieval examples.

The distinctive characteristics of damage here are entirely consistent with loading and destruction caused by seismic tremors and, as such, El Castillejo can be seen as an extraordinary material catalogue of the so-called Earthquake Archaeological Effects (known in the specialistic literature as EAE [6] or potential-EAE [5]). No other explanation can be convincingly supported by the geomorphological and archaeological evidence. First, landsliding appears unlikely: the buildings of El Castillejo were constructed on solid *in-situ* bedrock and the hillslopes show no signs of slope failure. Second, destruction by warfare is not supported by the archaeological record: no weapons were recovered by the archaeological investigations, and the fire identified in Trench 1 is limited only to this part of the hilltop, at least according to our present understanding of what is an extensively investigated archaeological site.

The evidence for fire in Trench 1 is also significant as it implies that hearths, braziers or candles at El Castillejo were alight when the destructive event occurred, so the village must have been inhabited at the time of the earthquake. The damage caused by the earthquake affected the 'living surface' [55], disrupted the settlement, and triggered a stratigraphic marker in the form of destruction layers. The collapse of buildings and the fires which ensued would have been all the more hazardous because of the density of domestic housing and restricted access down steep slopes. The absence of victims may perhaps be due to rescue efforts [56] but it also seems possible that severe foreshocks anticipating the final destructive main shock provided an opportunity for the villagers to leave. Whatever the case, the whole inhabited area was affected, not just individual buildings. As aforementioned, the site was reoccupied at a later stage, possibly soon after the occurrence of the seismic event. However, the village was not fully rebuilt to its pre-earthquake configuration; instead, only a few buildings were repaired employing low-quality and reused materials. Our trenches did not show any evidence of people resettling in the investigated sectors of the sites, and overall, the nature and purpose of this reoccupation remain unclear. We hope that further archaeological research can shed new light on the post-disaster phase of El Castillejo.

The 13th century earthquake which affected El Castillejo is not currently listed in seismic catalogues; indeed, no historic earthquakes are known at all in the Granada region prior to CE 1431 [26, 27] (Fig 8A). There are several possible reasons for this. It could be argued that many of the best studied buildings in this part of Andalusia post-date the earthquake. We think it

more likely however that the local site conditions at El Castillejo had a significant influence on the outcome. Steep topography, especially the crests of hills, can amplify ground-shaking [57] and, as we have seen, the terracing of the slope to accommodate housing may have enhanced the physical impact of the earthquake.

These effects would then have been further compounded by the choice of building materials at El Castillejo. In modern engineering studies rammed earth is usually classified as having high seismic vulnerability [58–62]. Although rammed earth walls are massive and heavy, their foundations are not deep and their unanchored corners are vulnerable to ground shaking. No matter how solid their composition, rammed earth blocks are inflexible and unable to mitigate seismic energy. The nearby medieval tower of the Castillo de Lojuela in Murchas, also constructed of rammed earth, has been shown to be vulnerable to 'very heavy damage' at earthquake intensities higher than IX (9.2). The expected PGA (peak ground acceleration) there is around 0.357g [62]. Translating this for the El Castillejo case study would generate a magnitude of higher than M5.5 ± 0.5, identical to the values calculated from our analysis of the archaeoseismic damage to structures on the site and the so-called 'site effect' [7, 63]. Based on these conclusions, there must be a causative fault in the vicinity of El Castillejo capable of producing an earthquake of c.M6.0.

## 4.2. The possible causative fault

The broad collision zone of the Eurasian and Nubian tectonic plates in southern Spain and especially Andalusia accommodates a NW-SE convergence of up to c.5 mm/year [64]. El Castillejo lies in the metamorphic Internal Zone of the Betic Cordilleras (see summary in Weijermars [65]), mainly consisting of marbles and schists which are mostly stable for construction purposes. Recent seismicity concentrates to the north in the Miocene post-orogenic Granada Basin and its bounding normal faults and to the west of El Castillejo along the boundary of the Internal and External Zones of the Betics [66, 67]. This is where seismicity has been documented historically, for example on 24 April 1431 at Atarfe near Granada or on 25 December 1884 in Zafarraya/Arenas del Rey [22, 68]. Both these earthquakes reached macroseismic intensities of VIII-IX (EMS), corresponding to magnitudes $M_S$ of 6.5 ± 0.2. Other reported historical events are smaller, the Albolote earthquake April 19 1956 and the Jayena earthquake June 24, 1984 reached M5 (EMS V-VII; [69]), at the threshold of causing physical damage in urban environments and sparsely settled areas. We suggest that three active tectonic faults are capable of generating the damage observed at El Castillejo [17]. The first candidate is the Ventas de Zafarraya Fault which has ruptured three times in the last c. 9,000 years with earthquakes significantly higher than M6, the last occasion being in 1884. With a length of c.15 km, this fault could be responsible for earthquakes up to M6.7 ± 0.3. However, the penultimate event occurred c.2,315 ± 30 years BP [23]. A second candidate is the Nigüelas (7 km long) and Padul Faults (8 km long; Gil et al. [70], both of which could generate earthquakes of magnitude M6.0–6.3 ±0.3 [16]. If both of these faults were to rupture at the same time, which is possible [70], an earthquake of magnitude M6.2 ±0.5 is likely. Other possible faults in the area and within the Granada depression are the Dílar Fault, which parallels the Padul Fault, but if this fault is mapped correctly, it seems too short to generate a M6.5 earthquake, and the longer La Mahalá Fault (15 km long), where a surface rupture displaced c. 887 ±48 years BP old deposits. Given that both have identical strikes, one hypothesis is that the La Mahalá Fault may connect to the Padul Fault and so form a much longer fault [71]. A fourth fault (the Albuñelas Fault) was excluded as it is too short (6.5 km max.) and no earthquake activity is noted in the QAFI database [17].

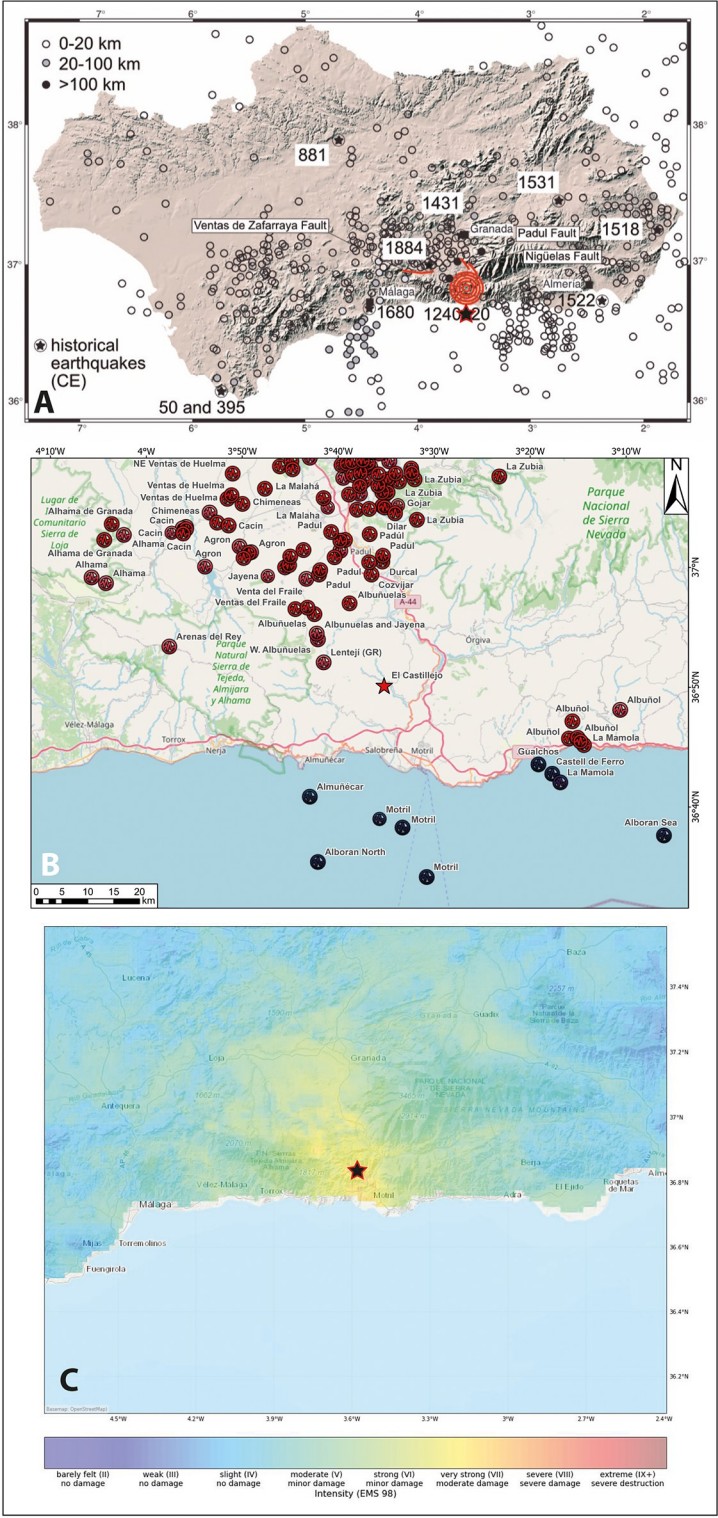

**Fig 8.** A: Seismicity in Andalusia, the red area shows the archaeological site at El Castillejo, the red lines are possible active faults in the vicinity (generated by K. Reicherter using QGis 3.26; Basemap: MDT200 2015 CC-BY 4.0 ign.es; historical earthquakes from [23], faults from the QAFI database [17]) B: recent earthquakes around El Castellejo (star) (K. Reicherter's own compilation; Basemap Open Street Map); C: Palaeo-shake map modelled for the missing El Castellejo earthquake, based on M6 and 10 km hypocentral depth [35] (generated using the ShakeMap map code by Jens Skapski; Basemap Open Street Map [http://usgs.github.io/shakemap/index.html]).

It is also possible that the earthquake which affected El Castillejo was caused by an as yet undiscovered active fault. This seems very unlikely because of the long history of seismic quiescence in the vicinity of the site. We tested hypothetical earthquakes and the isoseismal distribution of model earthquakes based on the damage observed at El Castillejo against known fault parameters and it seems most likely that the Nigüelas-Padul Fault System triggered the earthquake (Fig 8B). Unfortunately, like other faults in the vicinity of Granada, this fault lacks palaeoseismic investigation and any detailed assessment of the current seismic hazard (e.g. [71, 72]).

### 4.3. The recovery of a lost disaster

While El Castillejo lies in what might be described as a 'seismic gap' within the wider region of the Granada Basin and Internal Betic Cordilleras [22] (Fig 8C), it also sits in something of an historical 'black hole'. Geographically, its location on the western outreaches of the Sierra de Tejeda is remote and isolated. When the Christian kingdom of Castile occupied the Guadalquivir valley, conquering Córdoba (1236), Jaén (1246) and Sevilla (1248) during the 13th century, El Castillejo remained under the authority of a new Islamic dynasty, the Nasrids, whose emirate in Granada would endure until 1492. The paucity of written records for this period is due both to the less systematic conservation of documentary sources under the Nasrids [73] and later archival catastrophes perpetuated by the new Christian rulers: thousands of Arabic manuscripts were publicly destroyed in the Plaza Bib-Rambla in Granada at the end of the 15th century and many accounts of earthquakes and other natural disasters were presumably lost at that time [74, 75]. By contrast, from a region under Christian authority, letters of financial support survive in the Alicante area in the aftermath of the CE 1258 Onteniente earthquake [23, 76]. For geographical, geopolitical and historical reasons any written testimony of the El Castillejo earthquake was probably lost, if indeed it was ever produced in the first place.

## 5. Conclusions

The combined analysis of standing architecture and buried stratigraphy suggests that an earthquake struck the village at El Castillejo (Los Guájares, Granada) in the middle of the 13th century. Eight radiocarbon samples extracted from sealed archaeological layers provide consistent dates for a severe event of c.M6 ± 0.5 between 1224–1266 CE (or CE 1245±21) at 95.4% probability. This unknown event has not so far been captured by national or European seismic catalogues and currently represents the oldest historic earthquake in the Granada area.

In regions or historical periods in which documentary sources are scarce, archaeological recording and well-targeted investigation have the capacity to play an important role in the identification and dating of lost seismic disasters and thereafter in the refinement of historic earthquake catalogues. Our hope now is that other contemporary sites in the region can be investigated so as to understand the impact on the wider medieval landscape of al-Andalus. We anticipate that the results generated by this archaeological research might be captured by national and European seismic agencies and so have an impact on seismic disaster prevention and reduction.

## Supporting information

**S1 Fig.**
(JPG)

**S1 File. Radiocarbon dating.**
(PDF)

**S2 File. OxCal code.**
(DOCX)

**S3 File. PaleoShake maps (generated by K. Reicherter using QGis 3.26 [http://usgs.github.io/shakemap/index.html] and ShakeMaps 4 [http://usgs.github.io/shakemap/index.html]).**
(PDF)

**S4 File. Database and .klm file for contemporary earthquakes in Andalusia.**
(ZIP)

**S5 File. OSL dating.**
(DOCX)

# Acknowledgments

The authors are indebted to all those who participated in our fieldwork: Peter Brown, Elena Fiorin, Edward Treasure, Moisés Alonso Valladares, Juan Manuel Ríos Jiménez and Juan Antonio Rojas Cáceres. The map in Fig 1 was produced by Juan Antonio Rojas Cáceres and the identification of the radiocarbon-dated plant remains was carried out by Edward Treasure. We thank Andrew Millard (Durham University) and Giacomo Capuzzo (University of Trento) for their help with the Bayesian statistics of the radiocarbon dates obtained. We are very grateful to Jens Skapski (RWTH Aachen University) for his help in the elaboration of ShakeMap scenarios. We also thank the Municipality of Los Guájares and the Delegación de Cultura de la Junta de Andalucía, in particular Juan Cañavate Toribio, for supporting our research and supplying the requested permits. A special thank you is due to the welcoming residents of Guájar Faragüit. Finally, we thank the reviewers for their valuable feedback and comments that have greatly improved the quality of this work.

# Author Contributions

**Conceptualization:** Paolo Forlin, Klaus Reicherter, Christopher M. Gerrard, Alberto García Porras.

**Data curation:** Paolo Forlin, Klaus Reicherter, Christopher M. Gerrard, Ian Bailiff, Alberto García Porras.

**Formal analysis:** Paolo Forlin, Klaus Reicherter, Christopher M. Gerrard, Ian Bailiff, Alberto García Porras.

**Funding acquisition:** Paolo Forlin.

**Investigation:** Paolo Forlin, Klaus Reicherter, Christopher M. Gerrard, Alberto García Porras.

**Methodology:** Paolo Forlin, Klaus Reicherter, Christopher M. Gerrard, Ian Bailiff, Alberto García Porras.

**Project administration:** Paolo Forlin, Alberto García Porras.

**Resources:** Paolo Forlin, Alberto García Porras.

**Software:** Klaus Reicherter.

**Writing – original draft:** Paolo Forlin, Klaus Reicherter.

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
