## [Decision Letter · Decision Letter 0]

13 Jul 2023

PONE-D-23-05030Recovering a ‘lost’ seismic disaster. The destruction of El Castillejo and the discovery of the earliest historic earthquake affecting the Granada region (Spain)PLOS ONE

Dear Dr. Forlin,

Thank you for submitting your manuscript to PLOS ONE. After careful consideration, we feel that it has merit but does not fully meet PLOS ONE’s publication criteria as it currently stands. Therefore, we invite you to submit a revised version of the manuscript that addresses the points raised during the review process.

We look forward to receiving your revised manuscript.

Kind regards,

Jianhong Zhou

Staff Editor

PLOS ONE

Journal Requirements:

2. In your manuscript, please provide additional information regarding the specimens used in your study. Ensure that you have reported specimen numbers and complete repository information, including museum name and geographic location.

For more information on PLOS ONE's requirements for paleontology and archeology research, see https://journals.plos.org/plosone/s/submission-guidelines#loc-paleontology-and-archaeology-research.

"This study was initially conducted as part of the EU-funded ArMedEa project (‘The archaeology of Earthquakes in medieval Europe’, Marie Skłodowska Curie Action grant n.626659) and later supported by the British Academy (SRG/171316, ‘Dating an unrecorded medieval destructive earthquake: archaeological investigation at El Castillejo, Granada, Spain’) who funded the archaeological excavation and the analyses discussed here. Fieldwork at El Castillejo also formed part of the RiskRes project (‘Risk and resilience: historic responses to earthquakes in Europe, AD 1200-1755’) and we are grateful for the support provided by the Leverhulme Trust."

"PF was awarded a Marie Skłodowska Curie fellowship called ArMedEa (‘The archaeology of Earthquakes in medieval Europe’, grant n.626659) and a British Academy small grant (SRG/171316).

6. We note that Figures 1 and 8 in your submission contain map images which may be copyrighted. All PLOS content is published under the Creative Commons Attribution License (CC BY 4.0), which means that the manuscript, images, and Supporting Information files will be freely available online, and any third party is permitted to access, download, copy, distribute, and use these materials in any way, even commercially, with proper attribution. For these reasons, we cannot publish previously copyrighted maps or satellite images created using proprietary data, such as Google software (Google Maps, Street View, and Earth). For more information, see our copyright guidelines: http://journals.plos.org/plosone/s/licenses-and-copyright.

     1. You may seek permission from the original copyright holder of Figures 1 and 8 to publish the content specifically under the CC BY 4.0 license. 

In the figure caption of the copyrighted figure, please include the following text: “Reprinted from [ref] under a CC BY license, with permission from [name of publisher], original copyright [original copyright year].

Reviewers' comments:

Reviewer's Responses to Questions

**Comments to the Author**

1. Is the manuscript technically sound, and do the data support the conclusions?

Reviewer #1: Yes

Reviewer #2: Yes

2. Has the statistical analysis been performed appropriately and rigorously? 

Reviewer #1: Yes

Reviewer #2: Yes

3. Have the authors made all data underlying the findings in their manuscript fully available?

Reviewer #1: Yes

Reviewer #2: Yes

4. Is the manuscript presented in an intelligible fashion and written in standard English?

Reviewer #1: Yes

Reviewer #2: Yes

5. Review Comments to the Author

Reviewer #1: I am really glad to have received this manuscript for review. I find the subject matter very interesting. I would like to congratulate the authors for the interesting work and for providing new valuable historical and geological data. The authors explain the problem very well and discuss it in detail. I agree with their conclusions, and I think it should be published, but there are several points that I think need clarification before publication. I add here some suggestions to improve those aspects of the manuscript, so that it is clearer, as I think they are not well understood in the current text.

1. Questions about OSL dating

In the present format I do not understand the meaning of ages A and B of sample 438-2, and their relation with sample 438-5 (table 2 and lines 415-427. Please, explain better samples A and B. What is the meaning/interpretation of authors. After reading the text I understand that A represents a brick from a construction moment.

They also provide the overdispersion value for two of the three samples. Why not the other one?

They refer to the overdispersion value. Do they mean the overdispersion of the Central Age Model (Galbraith et al., 1999)?

If they are bricks, what does the overdispersion provide? This parameter is used to check complete bleaching of sediments, but bricks were fired. What it does provide?

In the Supplementary OSL material, the authors refer to sediment samples from three locations. I am, unable to see the reults.

2. Questions about the chronological model.

The authors used Oxcal and radiocarbon ages to get a more precise age. I agree, but they have other ages that could be used to improve the model. I refer to OSL ages.

Why the authors did not used such ages in the model?

Reviewer #2: This is an interesting paper for different reasons. Firstly, the paper argues the relevance of an undocumented event allowing to revise a previous stratigraphical sequence; Secondly, it provides a relevant approach to consider stratigraphical and architectonical formation processes analyzing structural deformation; Thirdly, the chronological strategy and the results are very cohesive with the site history; Finally, the paper puts the case study in a wider context.

There are only a few objections. In my opinion, it is needed a more detail explanation about the "extensive" excavation carry out in the 80s: surface, significance of the sample. It is strange that cracks are not attested in the east area of the site, and there are relevant differences in the external wall. What is more the distribution of the different deformations attested in figure 3 seems follow some kind of pattern.

I would like to know more about the reuse of the site after the seism event. Is there a hiatus? The reconstruction affects all the site? Is attested some change in the building techniques or materials? I believe that the cultural consequences of the event are not fully explored.

In a more general way, is this the first time that this approach has been used?

Finally, check the caption of supplementary material number 3.

6. PLOS authors have the option to publish the peer review history of their article (what does this mean?). If published, this will include your full peer review and any attached files.

Reviewer #1: No

Reviewer #2: **Yes: **Juan A Quirós Castillo

---

## [Author Response · Author response to Decision Letter 0]

15 Sep 2023

Responses to reviewers are included in the dedicated word file

---

## [Decision Letter · Decision Letter 1]

26 Feb 2024

PONE-D-23-05030R1Recovering a lost seismic disaster. The destruction of El Castillejo and the discovery of the earliest historic earthquake affecting the Granada region (Spain)PLOS ONE

Dear Dr. Forlin,

Thank you for submitting your manuscript to PLOS ONE. After careful consideration, we feel that it has merit but does not fully meet PLOS ONE’s publication criteria as it currently stands. Therefore, we invite you to submit a revised version of the manuscript that addresses the points raised during the review process.

The following paper is a well-supported study that will add to our knowledge of the history of the Iberian Peninsula. Both reviewers recommended it be accepted for publication - however one suggested a small correction regarding the size of the library. Therefore, I am returning the paper to you for you to consider this correction prior to full acceptance and publication of the paper in Plos One.

We look forward to receiving your revised manuscript.

Kind regards,

Joe Uziel

Academic Editor

PLOS ONE

Journal Requirements:

Reviewers' comments:

Reviewer's Responses to Questions

**Comments to the Author**

1. If the authors have adequately addressed your comments raised in a previous round of review and you feel that this manuscript is now acceptable for publication, you may indicate that here to bypass the “Comments to the Author” section, enter your conflict of interest statement in the “Confidential to Editor” section, and submit your "Accept" recommendation.

Reviewer #3: (No Response)

Reviewer #4: All comments have been addressed

2. Is the manuscript technically sound, and do the data support the conclusions?

Reviewer #3: Yes

Reviewer #4: Yes

3. Has the statistical analysis been performed appropriately and rigorously? 

Reviewer #3: N/A

Reviewer #4: Yes

4. Have the authors made all data underlying the findings in their manuscript fully available?

Reviewer #3: Yes

Reviewer #4: Yes

5. Is the manuscript presented in an intelligible fashion and written in standard English?

Reviewer #3: Yes

Reviewer #4: Yes

6. Review Comments to the Author

Reviewer #3: The evidence provided by this article of a hitherto undocumented earthquake in Los Guajares during the early Nasrid period is well argued and convincing. It is an important addition to our knowledge of seismicity in the Iberian Peninsula,

In the part dealing with how rammed earth reacts to seismic activity other examples - if known - both in the Iberian Peninsula and North Africa could be mentioned, perhaps even a reference to the recent earthquake in southern Morocco.

In the historical discussion, the reference to 'two million books' being destroyed in Granada is an exaggeration that makes no sense for the size of libraries at the time and should be corrected: the extant sources mention at most between four - five thousand mss. See Memorial de la vida de Fray Francisco Jiménez de Cisneros, ed. Antonio de la Torre y del Cerro (Madrid: Centro de Estudios Históricos, 1913), p. 35.

Reviewer #4: The research carried out is relevant and rigorous, with a high interest in the subject, little known and studied until now. The scientific contribution to the knowledge of historical earthquakes in southern Spain, through archaeological methodology, architectural study and absolute dating, is very original, novel and of great scientific interest. In fact, it has made it possible to know and date a historical earthquake prior to those recorded in documents written in that area.

The choice of topic and study samples, methodology, premises and hypotheses are correct and well applied.

In addition to the results obtained, the interpretation of the seismic events at the archaeological site should be highlighted, since the first excavations had considered that the collapses of structures had been caused by abandonment and subsequent erosion. This interpretation is now being revised, and attributed to an earthquake not documented so far, through a solid argument well supported by the archaeological, stratigraphic and dating analyses (14C and OSL) carried out.

The authors have correctly clarified the doubts and questions raised by previous reviewers, providing a solid and coherent argumentation. Likewise, the handwriting corrections have visibly improved the manuscript.

In short, the research is of high scientific quality and deserves to be published in an appropriate medium.

7. PLOS authors have the option to publish the peer review history of their article (what does this mean?). If published, this will include your full peer review and any attached files.

Reviewer #3: No

Reviewer #4: **Yes: **José Avelino Gutiérrez González

---

## [Author Response · Author response to Decision Letter 1]

27 Feb 2024

Pls see the attached file 'response to reviewers'

---

## [Editor Report · Decision Letter 2]

29 Feb 2024

Recovering a lost seismic disaster. The destruction of El Castillejo and the discovery of the earliest historic earthquake affecting the Granada region (Spain)

PONE-D-23-05030R2

Dear Dr. Forlin,

We’re pleased to inform you that your manuscript has been judged scientifically suitable for publication and will be formally accepted for publication once it meets all outstanding technical requirements.

Kind regards,

Joe Uziel

Academic Editor

PLOS ONE
---

## [Editor Report · Acceptance letter]

26 Mar 2024

PONE-D-23-05030R2 

PLOS ONE

Dear Dr. Forlin, 

I'm pleased to inform you that your manuscript has been deemed suitable for publication in PLOS ONE. Congratulations! Your manuscript is now being handed over to our production team.

Kind regards, 

on behalf of

Dr. Joe Uziel 

Academic Editor

PLOS ONE